# Extracellular Self- and Non-Self DNA Involved in Damage Recognition in the Mistletoe Parasitism of Mesquite Trees

**DOI:** 10.3390/ijms25010457

**Published:** 2023-12-29

**Authors:** Claudia Marina López-García, César Alejandro Ávila-Hernández, Elizabeth Quintana-Rodríguez, Víctor Aguilar-Hernández, Nancy Edith Lozoya-Pérez, Mariana Atzhiry Rojas-Raya, Jorge Molina-Torres, Jesús Alfredo Araujo-León, Ligia Brito-Argáez, Avel Adolfo González-Sánchez, Enrique Ramírez-Chávez, Domancar Orona-Tamayo

**Affiliations:** 1Medio Ambiente y Biotecnología, Centro de Innovación Aplicada en Tecnologías Competitivas (CIATEC), León 37545, Guanajuato, Mexico; marinalopez2508@gmail.com (C.M.L.-G.);; 2Centro de Investigación y de Estudios Avanzados (CINVESTAV), Instituto Politécnico Nacional, Irapuato 36821, Guanajuato, Mexico; cesar.avila@cinvestav.mx (C.A.Á.-H.); mariana.rojas@cinvestav.mx (M.A.R.-R.); enrique.ramirez@cinvestav.mx (E.R.-C.); 3Unidad de Biología Integrativa, Centro de Investigación Científica de Yucatán (CICY), Mérida 97205, Yucatán, Mexicojalfredoaraujo@gmail.com (J.A.A.-L.);; 4Facultad de Ingeniería, Universidad Autónoma de Yucatán (UADY), Mérida 97203, Yucatán, Mexico

**Keywords:** parasitic plants, DAMPs, ROS, JA, antioxidant enzymes, polyphenols, flavonoids

## Abstract

*Psittacanthus calyculatus* parasitizes mesquite trees through a specialized structure called a haustorium, which, in the intrusive process, can cause cellular damage in the host tree and release DAMPs, such as ATP, sugars, RNA, and DNA. These are highly conserved molecules that primarily function as signals that trigger and activate the defense responses. In the present study, we generate extracellular DNA (exDNA) from mesquite (*P. laevigata*) tree leaves (self-exDNA) and *P. calyculatus* (non-self exDNA) mistletoe as DAMP sources to examine mesquite trees’ capacity to identify specific self or non-self exDNA. We determined that mesquite trees perceive self- and non-self exDNA with the synthesis of O^2•−^, H_2_O_2_, flavonoids, ROS-enzymes system, MAPKs activation, spatial concentrations of JA, SA, ABA, and CKs, and auxins. Our data indicate that self and non-self exDNA application differs in oxidative burst, JA signaling, MAPK gene expression, and scavenger systems. This is the first study to examine the molecular biochemistry effects in a host tree using exDNA sources derived from a mistletoe.

## 1. Introduction

*Prosopis laevigata* (mesquite) is a Mexican endemic tree belonging to the *Fabaceae* family, predominately found in the center of Mexico and the south of the United States [1,2,3]. The mesquite tree has ecological and economic significance, given that it promotes soil fertility by fixing N_2_ and increasing PO_4_^−3^, Fe^3+^, and C levels [4,5,6,7]. Additionally, its biomass is used as biofuel, and its resin is utilized for the manufacture of footwear [1]. Unfortunately, the mistletoe *Psittacanthus calyculatus* poses a threat to this tree [8] and is a photosynthetically active angiosperm that has lost its ability to anchor to the soil. It instead develops an endophytic root system (haustoria) that anchors it to the vascular bundle of mesquite branches to extract water, minerals, and other nutrients [9,10,11,12,13,14,15].

*P. calyculatus* mistletoe is an obligate hemiparasitic aerial plant whose nutrient consumption causes starvation, growth loss, disease predisposition, and, ultimately, death to the infected host tree [16,17]. The prevalence of this parasitic plant is based on the behavioral patterns of frugivorous birds who eat fruits and contribute to their long-distance dispersal and the colonization of new hosts [16]. While sufficient information exists regarding the pollination syndromes that *P. calyculatus* uses to attract and manipulate pollinators and visitors [18], the molecular mechanisms of the infection process remain unknown. Currently, it is well-known that seeds that birds deposit on branches can germinate, leading to the haustorium tissue formation, and the emergence of this intrusive organ is accompanied by high activity of cell-wall-degrading enzymes [19]. The enzymatic degradation of the host bark causes a pit on the surface, that functions as an anchor for the haustorium penetration. When the haustorium begins to penetrate the host branches, it causes mechanical damage and breaks the host tissue, and the friction of the penetration also causes damage to the haustorium tissue, which further causes cellular damage that releases different damage-associated molecular patterns (DAMPs) [20,21,22].

DAMPs are ubiquitous molecules that are mainly composed of cell-wall fragments and compartmentalized biomolecules, such as ATP, DNA, and RNA [20,23]. These molecules function outside of the cell as elicitors, triggering reactions that promote tissue repair and immunization in order to prevent potential infections [24]. Additionally, plants can detect peptidoglycans or chitin to distinguish between self and nonself fragments of the cell wall [20], self RNA from double-stranded RNA [25], and self-DNA from microbial DNA by perception of GC-rich islands in unmethylated CpG motifs [26,27]. Nevertheless, in the case of mistletoe parasitism, the host plant must distinguish its self-patterns from those of the other plant. Hence, Table 1 provides a summary of the physiological responses to plant DAMPs in photosynthetic organisms, such as microalgae, model plants, and species of agronomic interest. In that aspect, pioneer experiments where the authors used common bean-leaf extracts as a cocktail of DAMPs were initially used to evaluate the plant’s response to self and non-self molecular patterns [28,29]. Subsequently, other authors, isolated and used molecules individually to analyze the stress responses to cell-wall fragments [30], extracellular ATP (exATP) [31,32,33,34,35,36,37,38,39], NTPs [31,40], and extracellular DNA (exDNA) [27,41,42,43,44,45,46,47,48,49,50,51,52,53,54,55,56,57,58].

Multicellular organisms suffer different types of cellular damage, including infectious processes. Janeway’s classical model states that the immune system evolved to distinguish the infectious nonself from the noninfectious self [59]. However, in most environments, injury to the outer layers of an organism (the skin or gut epithelia in the case of mammals and the epidermis of leaves and roots in the case of plants) inevitably leads to infection [45]. Thus, organisms require an endogenous signaling pathway that enables them to perceive injury through DAMPs, mount adequate local and systemic responses to activate the immune system and achieve resistance [45,60]. Therefore, the response to self-patterns is characterized by a rapid increase in reactive oxygen species (ROS) and cytosolic Ca^2+^ levels, activation of mitogen-activated protein kinases (MAPKs) cascades, jasmonic acid (JA) signaling and antioxidant systems, which induce long-term resistance to pathogens and inhibits their growth (Table 1). Alternatively, the response to nonself plant patterns varies among species. For example, close species show similar responses to self-patterns, while distant species only induce innate immunity to pathogens and polyphenol accumulation (Table 1). In summary, the compilation of data in Table 1 derived from performed assays on simple photosynthetic organisms (algae) and higher plants is consistent with a high conservation of the response to DAMPS [20,61].

Many plants are vulnerable to attacks by parasitic plants because they lack the ability to perceive them, and relatively less attention has been paid to the host tree’s ability to identify damaged compounds, including RNA, self exDNA, and cellular debris. In that sense, it has been recently investigated how DAMPs trigger defense-inducing signals, such as ROS production or MAPK signaling, minutes after harm is perceived. In the present study, we analyze mesquite’s ability to distinguish its self- or non-self exDNA from that of a parasitic mistletoe, because it is one of the most researched DAMPs in plants that demonstrates species-specific responses. We obtained exDNA from *P. laevigata* and *P. calyculatus* by genomic DNA fragmentation. Then, we conducted field experiments in an endemic mesquite area where we sprayed mesquite leaves with self- or non-self exDNA solution at two different concentrations. Further, we quantified the of O^2•−^, H_2_O_2_, polyphenols, and flavonoids concentrations, as well as the ROS–enzymes system that includes superoxide dismutase (SOD), peroxidase (PEX), and catalase (CAT) activities. We also quantified phytohormones content, and the relative expression levels of MAPKs as biomarkers of the activation of general stress against distinct exDNA sources. The results from the analysis of those biomarkers show that mesquite trees are able to distinguish between self-exDNA and non-self exDNA and suggest DAMPs as a potential biochemical signal that may be involved in the immune response against the progress of the parasitism. Furthermore, this is the first study to propose using exDNA as a tool for searching for molecular components that could be involved in mistletoe parasitism.

**Table 1 ijms-25-00457-t001:** Reports on the physiological responses during DAMP-mediated recognition in plant and microalgae species.

Specie	DAMP Type	Time of Response	Responses
*Arabidopsis thaliana*	Self-leaf extract	20–60 min	Increase in JA level [28]
*Phaseolus lunatus*	Self-leaf extract	20–60 min	Decrease in secretion of EFN [28]
			Increase in JA level [28]
			Upregulation of JA-related genes [28]
			Downregulation photosynthesis genes [28]
*Solanum lycopersicum*	Self-leaf extract	20–60 min	Increase in JA level [28]
*Fragaria* sp.	Self-leaf extract	20–60 min	Increase in JA level [28]
*Sesamun indicum*	Self-leaf extract	20–60 min	Increase in JA level [28]
*Zea mays*	Self-leaf extract	20–60 min	Increase in JA level [28]
*Phaseolus vulgaris*	Bean-leaf homogenate	2 h	Increase in secretion of EFN [29]
			H_2_O_2_ accumulation [29]
		24 h	Resistance to pathogens [29]
	Non-bean-leaf homogenate	24 h	Resistance to pathogens [29]
*Nicotiana benthamiana*	*Cuscuta* homogenates	1 h	Increase in ET level [62]
*Solanum lycopersicum*	*Cuscuta* Cell-wall homogenates	1 h	Increase in ET levels [62]
			Increase in ET levels [62]
			Increase lignification in application site [30,63]
*Vitis vinifera*	Self-xyloglucan from cell walls	5 min	MAPKs activation [64]
		20 min	H_2_O_2_ accumulation [64]
		1 h	Overexpression of *PAL* and stillbene synthase (*STS*) genes [64]
*Arabidopsis thaliana*	eATP	0 h	O_2_^•−^and H_2_O_2_ accumulation [31,34]
			Activation of Ca^2+^ channels [34]
			Upregulation of *MPK3* [34]
		30 min	Upregulation of genes respond to MeJA [37]
			Upregulation of *RBOHD* [37]
		1–2 h	Upregulation of SA-related genes [37]
			Upregulation of *RBOHD* and *PAL* [31]
		12 h	Inhibition of vesicular trafficking [35]
		24 h	Activation of ET signaling [65]
			Reduction of cell viability [35]
		10 d	Reduction of *JAZ1* stability [37]
			Resistance to pathogens [37]
			Decrease in leaf area and root length [39]
	NPTs (ATP, GTP, CTP)	5 s	Elevation in cytosolic Ca^2+^ [40]
			O_2_^•−^ accumulation [31]
*Nicotiana tabacum*	eATP	3–5 d	Upregulation of SA-related genes [33]
			Resistance to pathogens [33]
*Phaseolus vulgaris*	eATP	2h	H_2_O_2_ accumulation [38]
			Increase in CAT and PPO activities [38]
			Increase in malondialdehyde content [38]
		4 h	H_2_O_2_ level restored [38]
			CAT activity restored [38]
*Solanum lycopersicum*	eATP	30 min	NO accumulation [32]
*Dimocarpus longan*	eATP	1–5 d	Polyphenols and flavonoids accumulation [36]
			Decrease in PPO activity and total sugar content [36]
*Populus euphratica*	eATP	30 min	Elevation of cytosolic Ca^2+^ [66]
			Reduction of cell viability [66]
		12 h	H_2_O_2_ and NO accumulation [66]
*Chlamydomonas reinhardtii*	Self-exDNA	168 h	Inhibition of root growth [49]
			Forming aggregates [49]
*Nannochloropsis gaditana*	Self-exDNA	168 h	Inhibition of root growth [49]
			Forming aggregates [49]
*Neochloris oleoabundans*	Self-exDNA	15–60 min	Increase in PEX activity [54]
			Polyphenols and lipids accumulation [54]
		24–48 h	Increase in Cks and GA level [54]
*Arabidopsis thaliana*	Self-exDNA	15 min	H_2_O_2_ accumulation [58]
		1 h	Increase in JA level [56,58]
		2 h	Upregulation of *MPK3*, *MPK6*, *OXI1*, and *CML37* [47]
		8–10 h	Upregulation *RBOHF*, *LOX3*, *MYC2*, and *JAZ1* [56]
			cAMP production [53]
			Accumulation of RNA constituents [53]
		16 h	Increase in root hair density [48]
			Upregulation of CK’s-related genes [48]
		24 h	Downregulation ABA-related genes [48]
			Increase in SA level [58]
		5–10 d	Resistance to pathogens [47]
			Inhibition of root growth [56]
			Necrosis and chlorosis in root and leaf, respectively [48]
	Non-self exDNA (*A. thaliana* ecotypes)	15 min	H_2_O_2_ accumulation [58]
		1 h	Increase in JA level [56,58]
		5 d	Upregulation of *MPK3*, *MPK6*, *OXI1*, and *CML37* [47]
			Inhibition of root growth [56]
	Non-self- exDNA (*Arabidopsis pumila*)	1 h	H_2_O_2_ accumulation [56]
		5 d	Inhibition of root growth [56]
	Non-self exDNA (*Brassica*)	1 h	H_2_O_2_ production [56]
			Upregulation of *MPK3*, *MPK6*, *OXI1*, and *CML37* [47]
		24 h	Increase in SA level [58]
		5 d	Inhibition of root growth [56]
	Non-self exDNA (*Phaseolus vulgaris*)	1 h	Upregulation of *MPK3*, *MPK6*, *OXI1*, and *CML37* [47]
	Non-self exDNA (*Citrus aurantium*)	1 h	Upregulation *MPK3* and *OXI1* [47]
	Non-self exDNA (*Triticum aestivum* and *Solanum lycopersicum*)	1 h	No detectable effects [56]
	Synthetic ssODNs	1.5 h	Stomatal closure [46]
		4 h	Upregulation of *MPK3*, *PROPEP1*, and *WRKY33* [46]
		24 h	Resistance to pathogens [46]
*Acanthus mollis*	Self-exDNA	Nd	Inhibition of root growth [41]
	Non-self exDNA (*A. thaliana*, *Quercus ilex*)	Nd	No detectable effects [41]
*Phaseolus vulgaris*	Self-exDNA	30 min	H_2_O_2_ accumulation [45]
			Increase in JA level [45]
		12 h	MAPKs activation [45]
		24 h	Increase in SA level [27]
			Secretion of EFN [45]
			Resistance to pathogens [27,45]
		4 d	Reduction of herbivore [27]
		End growth cycle	Inhibition of root growth [45]
			Enhance in seed production [27]
	Non-self exDNA (*P. lunatus* and *Acacia farnesiana*)	24 h	Increase in SA level [27]
			Induction of resistance to pathogens [27]
	Self-exDNA	30 min	Increase of cytosolic Ca^2+^ [43]
	Non-self exDNA (*Zea mays*)	30 min	Increase of cytosolic Ca^2+^ [43]
		1 h	H_2_O_2_ accumulation [42,56]
			Polyphenols and flavonoids accumulation, CAT, SOD, and PAL activity [52]
			Inhibition of root growth [56]
		2 h	JA-Ile production [56]
		24 h	Upregulation of *PRRs*, *PIs ACC oxidase*, and calcium-related genes [42]
		48 h	Downregulation of *cat*, polygalacturonase, and coumaric acid to CoA genes [42]
			Increase in PAL activity [52]
		10 d	Upregulation *LoxD*, *MYC2*, *JAZ1*, and *RBOH* [56]
			Resistance to pathogens [56]
			Polyphenols and flavonoids accumulation [52]
	Non-self exDNA (*A. thaliana* and *T. aestivumand*)	1, 2, and 24 h	No detectable effects [56]
	Non-self exDNA (*Latuca sativa*, *Apium graveolens* and *Cucumis sativus*)	0–48 h	Increase in CAT activity, polyphenols, and flavonoids accumulation [52]
*Lactuca sativa*	Self exDNA	5 d	Upregulation of SOD, CAT and PAL [44]
			Polyphenols and flavonoids accumulation [44]
			Inhibition of germination and root growth [44]
	Non-self exDNA (*Capsicum chinense*)	5 d	Upregulation of SOD, CAT, and PAL [44]
			Polyphenols and flavonoids accumulation [44]
			Inhibition of germination and root growth [44]
	Non-self exDNA (*Acaciella angustissima*)	5 d	Upregulation of PAL [44]
			Polyphenols and flavonoids accumulation [44]
*Prunus persica*	Self-exDNA	12 h	Upregulation of MAPK-related genes [57]
			Increase in ethylene level [57]
			Upregulation of ET-related genes [57]
		36 h	Resistance to pathogens [57]
		5 d	Decrease in sugar content [57]
	Non-self exDNA (*S. lycopersicum*)	36 h	Resistance to pathogens [57]
		5 d	Decrease in sugar content [57]
*Alnus glutinosa*	Self-exDNA	72 h	Root damage [50]
	Non-self eDNA (*Festuca drymeja*)	72 h	Root damage [50]
*Oryza sativa*	Self-exDNA	7 d	O_2_^•−^ and H_2_O_2_ accumulation [55]
			Inhibition of root growth [55]
			Downregulation SOD and CA*T* [55]

ABA: abscisic acid, ACC: 1-aminocyclopropane-1-carboxylic acid, CAT: catalase, CKs: cytokinins *CML37: calmodulin-like 37*, EFN: extrafloral nectar, ET: ethylene, GA: gibberellin acid, JA: jasmonic acid, JA-Ile: jasmonoyl-isoleucine JAZ1: jasmonate-zim-domain protein 1, *LOX3: lipoxygenase 3*, *LoxD: linoleate 13S-lipoxygenase 3-1*, MeJA: methyl jasmonate, *MPK3: mitogen-activated protein kinase 3*, *MPK6: mitogen-activated protein kinase 6*, *MYC2: Basic helix-loop-helix (bHLH) DNA-binding family protein*, *OXI1: oxidative signal-inducible 1*, PAL: PHE ammonia lyase, SOD: Super-oxide, CAT: catalase, PEX: peroxidase, *PIs: proteinase inhibitors*, PPO: polyphenol oxidase, *PRRs: pattern-recognition receptors*, *PROPEP1: precursor of peptide 1*, *RBOHD: RESPIRATORY BURST OXIDASE HOMOLOG D*, *RBOHF: RESPIRATORY BURST OXIDASE HOMOLOG H*, SA: salicylic acid, SOD: superoxide dismutase, and *WRKY33: WRKY DNA-binding protein 33*.

## 2. Results

### 2.1. Study System

In order to evaluate mesquite’s ability to respond to specific self or nonself patterns, we obtained exDNA from *P. laevigata* (self-exDNA) and *P. calyculatus* (non-self exDNA) by genomic DNA sonication extracted from field-collected leaves. To bring the data as close as possible to what happens in nature, mesquite leaves under field conditions were sprayed with deionized water (as a control) as well as with self- and non-self exDNA solutions, and and mesquite leaves had been mechanically damaged with a metal bristle brush (as a positive control for cell damage). Sonication produces exDNA fragments of approximately 200–1200 bp that were applied onto the leaf surfaces. exDNA solutions were prepared at two concentrations that are known to be efficient in inducing responses comparable to those generated by other plant species, with 100 µg/mL applied to Arabidopsis, *Phaseolus vulgaris*, *Phaseolus lunatus*, *Lactuca sativa*, and *Capsicum annum* [44,47,56,57,58,67], and 200 µg/mL to Arabidopsis, *Phaseolus vulgaris* and *Phaseolus lunatus* [43,44,45,47,56]. Leaf samples were collected at 0, 1, and 2 h after the treatments, a consensus time at which there is an accumulation of ROS and activation of JA signaling in different plant species as mentioned before (Table 1). 

### 2.2. Effect of exDNA on the Oxidative Burst

The first line of defense of the innate immune system of plants coordinates defense responses through ROS, which are produced in the area where cellular damage has been detected due to the perception of DAMPs. In diverse reports, O_2_^•−^ is rapidly accumulated at the apoplast [68,69], but due to its instability and its toxic effects in tissues, it is converted to H_2_O_2_ [70,71]. Cell damage usually increases the ROS level in an interval of 30 to 60 min (Table 1). For this reason, we detected O_2_^•−^ and H_2_O_2_ using the NBT and xylenol orange method, respectively, at 0, 1, and 2 h after different source treatments.

In that aspect, O_2_^•−^ levels in mesquite leaves increased at different times (0, 1, and 2 h) with 100 or 200 µg/mL with self or non-self exDNA, respectively (Figure 1a). However, we detected O_2_^•−^ after self exDNA treatments, which increased significantly with the application of 200 µg/mL of self exDNA, and the O_2_^•−^ levels subsequently decreased at later times (1 and 2 h) (Figure 1a). In damage treatment, the O_2_^•−^ increased at one hour of measurement and decreased at 2 h (Figure 1a).

Regarding the accumulation of H_2_O_2_ levels with the exDNA source applications, in the control treatment, H_2_O_2_ levels were not detected at early or later times. In contrast, the H_2_O_2_-level increased in a high concentration immediately after damage treatment and decreased at subsequent times. On the other hand, H_2_O_2_ concentrations increased with the addition of 100 µg/mL of self exDNA at early times (0 h) and diminished dramatically at 1 and 2 h. In contrast, we observed that applications of self exDNA at 200 µg/mL, and non-self exDNA at 100 or 200 µg/mL did not raise H_2_O_2_ concentrations at early time (0 h). The treatment with non-self exDNA at 200 µg/mL resulted in an increase of H_2_O_2_ levels at later times (2 h) (Figure 1b). 

### 2.3. Accumulation of Polyphenols and Flavonoids

The phenylpropanoid pathway (PPP) is positively regulated by JA [72], which is the precursor of many families of compounds, such as phenylpropanoids, flavonoids, lignins, monolignols, phenolic acids, stilbenes, and coumarins, which participate in the defense response [73]. Particularly, mesquite stands out in the plant kingdom for producing many polyphenols and flavonoids [74]. Therefore, we quantified the level of these molecules with the exDNA source application using spectrophotometric techniques.

We detected slight upward variations in the polyphenol content with the different treatments. For example, mechanical damage and application of 100 and 200 µg/mL of non-self exDNA presented the highest levels of polyphenols at time 0. In contrast, 100 µg/mL of self exDNA induced polyphenol levels at 1 h, while non-self exDNA treatment did not induce polyphenol levels. On the other hand, at a later time (2 h), the polyphenol content remained constant with exDNA application, and only damage treatment increased their polyphenol levels (Figure 2a). 

Contrary to what was observed with polyphenols, flavonoids have shown significant differences in their content between conditions. All treatments, including mechanical damage, self-exDNA, and non-self exDNA applications, showed increases in the flavonoid content compared to the control. Wounded leaves increased the flavonoid content at the early and later times (0, 1, and 2 h). Similarly, 100 or 200 µg/mL of self exDNA, induced the flavonoid levels, and their concentration remains constant at later times. In a similar manner, 100 µg/mL of nonself exDNA showed a flavonoid content greater than damage during the first hour after treatment, but the content decreased after 2 h, and 200 µg/mL of nonself exDNA showed contents similar to the damage treatment (Figure 2b).

### 2.4. Antioxidant Enzymes

An increase in the ROS rate, as shown in Figure 1, causes damage to host tissues [70]. To avoid this damage and return to a redox balance, cells activate ROS scavenger systems [75,76]. Thus, polyphenols act as nonenzymatic scavengers during damage or exDNA recognition. Therefore, to complement the ROS system, we analyzed the enzymatic anti-ROS machinery composed of the differential activities of SOD, PEX, and CAT, a conserved antioxidant system that regulates the level of ROS in the recognition of DAMPs (Figure 3). The O_2_^•−^ is reduced to H_2_O_2_ by SOD, and H_2_O_2_ is decomposed into H_2_O and O_2_ by PEX and CAT [54,70,71]. SOD activity increases in wounded leaves at later times (2 h); however, the different exDNA applications only slightly increase in SOD levels (Figure 3a). PEX enzyme activity increased in response to damage and exDNA sources (Figure 3b); CAT activity increased when 100 and 200 µg/mL of non-self exDNA were applied at 0 h and decreased at 1 or 2 h. (Figure 3c).

### 2.5. Activation of JA Signaling

The oxidative burst brought on by necrotic pathogen infection, herbivore attack, and mechanical damage causes JA biosynthesis to occur within minutes [77,78]. The ROS found earlier and shown in Figure 1 may have increased the JA content. In our study system, the damage treatment is the only one that exhibits a statistically significant rise in JA levels within the first hour. Self-exDNA application at a concentration of 100 µg/mL did, however, show an increase within the same hour but not a statistically significant one. Comparable values were observed for the remaining treatments and the control conditions (Figure 4).

### 2.6. Hormonal Accumulation during the Perception of Self- and Non-Self exDNA

The existence of crosstalk between the signaling pathways of defense hormones, such as abscisic acid (ABA), JA, and salicylic acid (SA), with growth regulatory hormones, such as auxins and cytokinins (CKs), is known to modulate and coordinate defense responses [79,80]. In parallel to the quantification of JA, we quantified two other defense hormones (SA and ABA) and four canonical growth regulatory hormones by HPLC. We observed that the SA levels increased with 200 µg/mL self-exDNA at later times (2 h), and the non-self exDNA applications did not exert an increase in the concentration of this phytohormone (Figure 5a). The treatments did not show statistically significant differences for ABA levels at 0 h. However, after applying 100 µg/mL of self-exDNA for one hour, there was a noticeable rise in the amount of ABA (Figure 5b). 

Additionally, we evaluated three cytokinins (CKs) that are related to leaf development. For example, zeatin was not found following various treatments, and iP did not exhibit statistically significant changes at any exDNA source application (0, 1, or 2 h), except for 100 µg/mL of non-self exDNA application, which caused an increase in iP at 2 h (Figure 5c). Conversely, Kinetin levels were detected (1 or 2 h) after applying 100 or 200 µg/mL of non-self exDNA and self-exDNA (Figure 5d). 

To coordinate the many plant-development processes, auxins are generated in the shoot system and then transferred to the other organs via the phloem [81]. In host and parasite plants, auxins mainly participate in the development of the root system and haustoria, respectively [82,83,84]. Finally, we determined that the IAA phytohormone was present in the early stages of mechanical damage (0 or 1 h), but those levels were not detectable when exDNA treatments were applied. In contrast, after two hours of treatment, there was a rise in non-self exDNA at 100 or 200 µg/mL (Figure 5e). 

### 2.7. Mapk Gene Expression Levels by Self and Non-Self exDNA

MAPK pathways are common and versatile signaling proteins that lie downstream of second messengers and play central roles in plant responses to various stresses [85] such as wounding, pathogen recognition, and general defenses [86]. To determine whether the exDNA applications activate *MAPK* genes, we examined the relative expression of three MAPK gene family members using RT-PCR. *MAPK2*, *MAPK4*4, and *MAPK10* genes are associated with mechanical damage, immune defenses, and general stresses like plant pathogens. Accordingly, following mechanical damage, the activation of the *MAPK2* gene expression was detectable at an early time (0 h), increased at 1 h, and gradually decreased at 2 h (Figure 6a). Furthermore, *MAPK2* levels increase with the application of 200 µg/mL of self-exDNA and 100 and 200 µg/mL of non-self exDNA at 0 h. However, gene expression increases with any exDNA concentration at late times (1 or 2 h). Additionally, when 200 µg/mL of self-exDNA and 100 and 200 µg/mL of non-self exDNA are applied at 0 h, *MAPK2* increases. Moreover, at late times (1 or 2 h), gene expression increases with any exDNA concentration. Regarding the *MAPK4* gene, our findings show that 100 µg/mL of non-self exDNA application increased the relative expression at 0 h, and that this expression decreased at later times (1 or 2 h). Additionally, we observed that the gene expression always remained stable, except for the 100 µg/mL of self-exDNA application that decreased at 2 h (Figure 6b). However, early time (0 h), 100 and 200 µg/mL of self-exDNA increased *MAPK10* gene expression significantly compared to 100 µg/mL of non-self exDNA. Nevertheless, those expression patterns decreased later (1 or 2 h) (Figure 6c).

## 3. Discussion

Based on the different perceptions of the DAMPs application (in particular exDNA), and the plant immune responses, we investigated distinct immunity-related traits that were similar to those reported in other plant species (Table 1) by using self-exDNA applications from *P. laevigata* (mesquite) and non-self exDNA from *P. calyculaus* (mistletoe), which are phylogenetically distant from each other and belong to different families (Fabaceae and Loranthaceae, respectively) [87,88]. Data obtained in this study indicated that O_2_^•−^, H_2_O_2_, flavonoids, and ROS-stress enzymes were triggered; additionally, phytohormones and MAPK gene expressions were accumulated spatially, regardless of the self-exDNA or non-self exDNA applications.

Plants’ innate immune systems use ROS to initiate defense responses when they detect cellular damage as a result of detecting foreign materials (DAMPs) [69,89]. O_2_^•−^ is accumulated inside the cell apoplast, but, due to its instability and its toxic effects on tissues, it transforms into H_2_O_2_ [69,70]. Our observations indicate that O_2_^•−^ and H_2_O_2_ levels were elevated at early times (0 or 1 h) in response to 100 or 200 µg/mL of self- and non-self exDNA. Biomarkers of oxidative burst with the exDNA source treatments suggest that the self-exDNA application could be an indicator of the stress levels similar to the damage suffered in mesquite tissues. In that aspect, similar effects were reported with the application of self-exDNA to root rice, which increased the production of O_2_^•−^ and H_2_O_2_ levels [55]. Moreover, Vega-Muñoz et al. [58] and Zhou et al. [56] discovered that there were significant levels of H_2_O_2_ in both self- and non-self exDNA applications in Arabidopsis ecotypes at early times. In addition, *Solanum lycopersicum* [42,56], *Brassica napus* [56], and *Phaseolus vulgaris* plants [45] showed high H_2_O_2_ levels with the self-exDNA application. Taken together, self- and non-self exDNA can induce a species-specific response such as ROS in mesquite trees.

Specifically, large concentrations of metabolite accumulations, such as flavonoids and polyphenols were observed by the exDNA applications. For example, we saw that polyphenols were present at detectable quantities in all assessed conditions, regardless of whether the exDNA applications were made early or late. Flavonoid concentrations were produced in trees that underwent mechanical damage or non-self exDNA treatments. *Prosopis* leaves contain a significant amount of several metabolites, including flavonoids and polyphenols, and it has been determined that total polyphenols in *P. glandulosa* and *P. juliflora* trees exposed to metal stress are constitutively present [90,91]. This could potentially account for the observed increase in polyphenol concentration following any kind of treatment, such as mechanical injury or exposure to exDNA sources. Recently it has been observed that the use of either self- or non-self exDNA in herbaceous plants has a significant effect on the amount of polyphenols and flavonoids. For example, in *L. sativa* and *S. lycopersicum,* the presence of self-exDNA or non-self exDNA increased the total polyphenols and flavonoids [44,52]. To our knowledge, non-self exDNA applications strongly induce the synthesis of flavonoids, which serve as an auxiliary antioxidant system for host damage, with a specific function of diminishing ROS accumulation in plant cells [92]. However, to avoid the stressors, the plants can use antioxidant enzyme systems to maintain and regulate their own homeostasis. In that aspect, SOD, PEX, and CAT enzymes revealed differential activities with non-self exDNA sources. SOD and PEX activities were induced with both exDNA sources; in contrast, CAT activities were induced only with non-self exDNA. These observations are similar to those obtained in tomato plants, where the application of self and non-self exDNA induced the enzymatic activities of SOD and CAT [52]. In the same way, in *L. Sativa*, the applications of self and non-self exDNA induced the differential expression of genes-encoding SOD- and CAT-like enzymes [44]. Those results might suggest that self and non-self-exDNA induce the activation of the enzymatic antioxidant system necessary to decrease the intensity of ROS. 

It is well known that the biosynthetic pathways of plant phytohormones involve the interaction of growth-regulating hormones, such as auxins and cytokinins (CKs), and defense hormones, such as salicylic acid (SA), JA, and abscisic acid (ABA), which play important roles as primary messengers in signal transduction and cell-metabolism regulation [91]. Additionally, they synchronize and influence defense responses through signaling pathways under different plant stresses [79,80]. In this regard, plant research to detect phytohormones after applying exDNA sources has been based mainly on JA and SA analysis. Under mechanical damage and after applying 100 µg/mL of self-exDNA, we found a rise in JA levels associated with an oxidative burst; however, non-self exDNA treatment did not affect JA levels. Additionally, JA levels were found with the application of exDNA sources in a similar way to the different reports mentioned [25,32,36,55,57,87]. Additionally, we quantified SA and ABA, and four canonical growth regulatory hormones such as Zeatin, Kinetin, iP, and IAA derived from the self and non-self exDNA applications.

SA is a signal molecule that is derived from the Shikimate–PPP pathway in plants and is recognized for its role in antioxidant defense systems and gene regulation [93]. Moreover, the synthesis and metabolism of SA are necessary for the development of the stress symptoms and hypersensitive response, as well as the induction of gene-encoding pathogenesis-related proteins [94]. We observed a pattern of SA contents in both self and non-self exDNA treatments, with high amounts at early and late times. In that aspect, many authors have reported that SA peaked minutes after or at 24 h after DAMP sources, similar to our results reported here [25,32,36,55,57,87]. Notably, we detected an increase in ABA contents with non-self exDNA treatment at 2 h after treatment. ABA and JA signaling pathways converge at several regulatory points that are involved in ROS activation, *MYC2* upregulation, and stomatal closure [78,95,96,97,98,99]. However, these pathways are erroneously considered to be antagonistic, and it is also speculated that biotic stresses like DAMP sources do not cause the ABA pathway to be activated. Chiusano et al. [48] analyzed ABA signaling with self and non-self exDNA applications, and they observed a downregulation of ABA-related genes with self-exDNA application at later times. Auxin and CKs have great potential to mitigate a variety of stressors and control plant development and stress tolerance (Raza et al., 2023). For example, CK (6-benzylaminopurine; BAP) and auxin (N6-(2-isopentenyl)adenine; iP) levels increased in *A. thaliana* at 16 and 24 h after treatment with self-exDNA [48]. Additionally, an upregulation of CK-related genes was detected by the exDNA source applications [48]. Similar effects were observed in *Neochloris oleabundans* microalgae with the self-exDNA applications, and differential responses in gibberellic acid, isopentenyladenine, and benzylaminopurine phytohormones compared with non-self exDNA application [54]. In that aspect, we detected an increase in iP and Kinetin levels with self-exDNA and non-self exDNA treatment at 1 or 2 h. Subsequently, we observed a differential auxin content with 200 µg/mL of non-self exDNA at a late time. These data could indicate that the differential synthesis of CKs and auxin are regulated by the indistinct exDNA source application.

Plants require an endogenous signaling pathway that enables them to perceive danger, for example signals that can originate from the infection and injury through DAMPs to mount adequate local and systemic responses to activate the immune system and achieve resistance [100]. DAMPs that trigger an immune response are ATP, extracellular matrix fragments, and exDNA, which activate signaling minutes after the perception of damage with events such as calcium fluxes in the cytosol, ROS, and MAPKs, which are considered conserved early responses [101]. Hence, exDNA source application in *P. laevigata* activates stress-related responses. In that aspect, our observations indicate that mechanical damage, both self and non-self exDNA induce the expression of MAPK-related genes like *MAPK2*, *MAPK4*, and *MAPK10*. In particular, there was a slight increase of *MAPK2* relative expression with exDNA sources but a significant increase with mechanical damage at one hour. Applying non-self exDNA can increase the relative expression of the *MAPK4* gene at either early or late stages. Finally, the gene expression of *MAPK10* increases its relative expression exclusively at early times when using self exDNA. These findings suggest that signaling and regulation of self and non-self exDNA, which the tree perceives as a stressor, is the reason for the increase in the relative expression of various MAPK genes. This aspect of plant-stress perception may be explained by the specific role that each encoding gene for the MAPK examined plays in its expression. For example, *MAPK2* is a gene family member that is involved in mechanical plant wounding. After herbivory with *Helycoverpa armigera* and mechanical wounding treatments, the gene expression levels of *MAPK2* changed in soybean cultivars [102], indicating that herbivores activate this gene. By the way, *MAPK4* is a related gene in plant immune response against phytopathogens, due to mediate jasmonate- and salicylate-dependent defense responses, and acts with other MAPKs as a negative regulator of plant immune responses [103]. Additionally, the gene *MAPK10* is associated with resistance to plant pathogens [104]. Previous studies support our findings. For example, single-stranded, and exDNA source applications from Arabidopsis induce the early expression of different *MAPK* genes involved in plant defenses [46,47]. Moreover, the self exDNA application of *Phytophthora capsici* induces the relative expression of *MAPK1,* that are involved in the pathogenicity of the fungus in chili peppers [105]. Duran-Flores and Heil [45] found MAPK accumulation in bean leaves treated with mechanical damage and exDNA sources. Further, in peach fruits, a complete MAPK cascade (MAPKKK1, MAPKK2, and MAPK1) was identified, which functions downstream of the FLS2–BAK1 receptor complex to exDNA [57]. However, phosphorylation of MAPKs is one of the first reactions that cells have to DAMP sensing [106]. exDNA is recognized by surface proteins known as pattern-recognition receptors (PRRs), which sets off immune responses such as Ca^2+^ ion fluxes across the plasma membrane, the production of ROS, the activation of MAPKs, and a quicker regulation of defensive gene expression [19,24,41,43,47,58,99] to phosphorylate NADPH oxidases (NOX family) to produce O_2_^•−^ in the apoplast, initiating the oxidative burst within a few minutes [107]. Based on the above, the application of self-exDNA as a damage-specific DAMP strongly induces MAPK regulation in mesquite trees. 

Mechanical damage, herbivory, and pathogen attack release DAMPs such as cell wall fragments, eATP, exDNA, and RNA, DAMPs recognition is a highly conserved mechanism across kingdoms [61,108]. DAMPs can mediate the different plant immune responses. For example, they can cause an increase in the levels of H_2_O_2_ and JA, activities of PPO, CAT, PEX, SOD, and hundreds or thousands of genes change their expressions. In particular, those that encode MAPKs, pathogenesis-related (PR) proteins, and enzymes of phenylpropanoid metabolism, as well as components of the JA and ethylene pathways in short or large period times induce resistance to pathogens through salicylic acid (SA) and ethylene (ET) signaling, downregulating photosynthesis genes, reducing cell division, and suppressing leaf and root growth [28,33,44,52,54,56,58,109,110,111,112,113]. 

### Relationship of exDNA in the P. calyculatus Establishment

*P. calyculatus* has developed intelligent strategies to obtain nutrients from host trees through a structure called haustorium. The infective process takes place during seed germination, and the appearance of a prehaustorium occurs which activates the expression of genes encoding an arsenal of cell-wall-degrading enzymes that help to breach the host bark and the cortex [19]. Through mechanical pressure, the haustorium penetrates the branch and will connect with the xylem host [19]. The host experiences mechanical wounding during this penetration process, and the haustorium tissue may also experience mechanical wounding as a result of friction between the branch’s woody walls due to the narrow process inside the branch. Then, the release of cellular compounds (self or non-self) from both species may occur, and the DAMP perception processes occurs, triggering different stress responses that activate the cascade of defense responses in the host tree. Thus, organisms require an endogenous signaling pathway that enables them to perceive injury through DAMPs, mount adequate local and systemic responses to activate the immune system and achieve resistance such as MAPKs [100]. For example, in *Striga hermonthica*, a root parasite, the haustorium formation is induced by the haustorium-inducing factors secreted by host plants, such as Dimethoxybenzoquinone (DMBQ). DMBQ may induce the primary signaling cascade for haustorium induction through leucine-rich repeat-receptor-like kinase (CADL LRR-RLKs). It is known that DMBQ treatment of *P. japonicum* induces Ca^2+^ signaling and MAP kinase (MAPK) phosphorylation. However, many elements of the signaling cascade are unknown. This includes the coreceptor for the CADL LRR-RLKs, the receptor-like cytoplasmic kinase (RLCK), the specific MEKKK, MEKK, and MAPK signaling cascade involved, and the Ca^2+^ channel that facilitates Ca^2+^ signaling, which subsequently induces pattern-triggered immunity [114]. Tomato plants were able to respond to a small peptide factor found in the parasitic plant *C. reflexa*’s extracts because they contain a cell-surface-receptor-like protein *CUSCUTA RECEPTOR 1* (*CuRe1*), which is essential for the perception of this parasite-associated molecular pattern, as well as a rapid induction of ROS in a similar way to microbial pathogens [30]. Moreover, *C. campestris* parasite extracts were injected into tomato plant hosts, and the host plants responded with reinforced cell walls at the infection sites against the parasite, thus, avoiding the parasitic infestation [62,63]. These events may be related to lignocellulolytic degradation of the host’s cell wall, and the released fragments accumulate by mechanical injury or enzymatic degradation that may serve as DAMPs and trigger defensive responses in the host [115]. In addition, transcriptome analysis in the infective process of *Psittacanthus schiedeanus* mistletoe, an arsenal of cell-wall-degrading enzymes like glucosyl hydrolases is necessary for the host infection [116]. Similarly, enzymatic analyses in *P. calyculatus* mistletoe revealed high activities of cell-wall-degrading enzymes such as cellulase and β-1,4-glucosidase that were primarily active in haustorium development, while xylanase and endo-glucanase were highly active in the haustorium penetration and xylem connection [19]. Additionally, we found that the host reacts against *P. calyculatus* mistletoe by producing oxidative stress processes like H_2_O_2_, ROS-related enzymes, and different concentrations of phytohormones like JA and SA [117]. These findings suggest that the host perceives self or non-self DAMPs. However, it is difficult to identify DAMPs as molecular effectors at particular stages of the infectious process, because they cannot be distinguished from the species from which they originated. The term “DAMPs” refers to self and non-self patterns that may originate from parasitic or host plants. These patterns are understood to be a possibility in the plant–host interaction, but further research must be conducted to rule out this theory. Hence, the approaches that we have taken in this research show us that exDNA is perceived as a DAMP. That is relevant in the mistletoe–mesquite interaction, which is an area where it has not been explored indepth. That requires the application of state-of-the-art tools such as “-Omics” to exploit in the future, with results that open a window of new knowledge, and those can be applied to other models of parasitic interaction, such as mistletoes that that infect host trees and root-parasitic plants. 

Based on our results, we propose a working model of molecular interaction between the self and non-self exDNA host-tree perception (Figure 7). The application of self and parasite exDNA in mesquite leaves under field conditions allowed us to determine that mesquite recognizes and responds differentially to self- and non-self exDNA. exDNA is recognized as a DAMP by the cell host, activating MAPK signaling and increasing ROS, such as O_2_ and H_2_O_2_. Subsequently, the antioxidative system, including enzymes such as SOD, PEX, and CAT, will help reduce this oxidative burst. Additionally, flavonoids detoxify H_2_O_2_ by preventing signal amplification, and then JA levels increase. Surprisingly, there is an accumulation of flavonoids, which is independent of hormonal levels. After a period, H_2_O_2_ levels return to their initial levels, flavonoids do not change, and kinetin and ABA phytohormone increase, which may cause flavonoid accumulation or prevent JA signaling. Lastly, levels for phytohormones such as kinetin and ABA return to initial levels after 2 h, but flavonoid levels peak. On the contrary side, independent of hormone accumulation, nonself exDNA perception induces H_2_O_2_, and CAT activity detoxifies these ROS, and flavonoid accumulation. Later, there is a decrease in CAT enzyme activity, a rise in H_2_O_2_, and a buildup of other phytohormones such as IAA.

## 4. Materials and Methods

### 4.1. exDNA Generation

Fresh mesquite (*P. laevigata*) (Appendix A) and mistletoe (*P. calyculatus*) leaves were collected near Irapuato, Guanajuato (20°42′42.18″ N; 101°19′22.66″ O) and frozen with dry ice. To obtain genomic DNA, 5 g of ground tissue were homogenized with 20 mL of 100 mM Tris-HCl pH 8, 5 mM EDTA pH 8, 50 mM NaCl, and 10 mM β-mercaptoethanol [118]. Genomic DNA concentrations were determined on a μDrop™ Duo plate and quantified in a microplate spectrophotometer (Multiskan SkyHigh, Thermo scientific, Waltham, MA, USA), then the extractions were treated with RNase to remove contaminating RNA. A solution of DNA (600 µg/mL) was resuspended in ultrapure and sterile water. We obtained short fragments of exDNA between 200–1200 pb after DNA sonication with the following parameters: 6:00 min at 55% of amplitude with a pulse of 1 s (active) and 1 s (not active). The exDNA concentration generated was adjusted with sterile distilled water to reach 100 or 200 µg/mL in the final concentrations that are reported to be effective in other model plants. Genomic exDNA, as well as fragmented exDNA, was verified on 3.0% agarose gel using ethidium bromide staining (Appendix A).

### 4.2. Study Site and exDNA Application

Field experiments were performed in a suburban area near Irapuato, Guanajuato. Young leaves and branches from mesquite trees were selected without visible infection by pathogens or herbivores, and, then, the branches selected were tagged. 

Ten mL of self exDNA (*P. laevigata*) solution and nonself exDNA (*P. calyculatus*) at two concentrations (100 or 200 µg/mL with 0.1% of Tween 20) were sprayed onto the surface of the mesquite leaves from branches. Additionally, 10 mL of sterile deionized water was applied to the control branches. Simultaneously, other leaves from the branches were damaged with a steel-wire brush as a positive control for cell damage. Finally, the leaves were collected at 0, 1, and 2 h, and samples were flash-frozen with liquid nitrogen and stored at −80 °C until experimental use. 

### 4.3. Quantification of O_2_^•−^

The method can quantify O_2_^•−^ radicals that are produced in plant tissues under stress and is based on the selective extraction of formazan produced after reduction with nitroblue tetrazolium (NBT) in histochemical staining [119]. O_2_^•−^ was measured according to Fonseca-García et al. [120], with some modifications. Briefly, 500 µL of 0.1% NBT (Sigma Aldrich, St. Louis, MO, USA) in 50 mM pH 7.5 NaH_2_PO_4_ were added to a tube containing 10 pinnae (leaves). Samples were incubated at room temperature for 1 h in darkness. Then, NBT-stained leaves were bleached with 96% ethanol incubated for 1 h [121]. NBT-stained leaves were ground with liquid nitrogen and dissolved in 2 M KOH-DMSO (1:1.6 *v*/*v*) and then centrifuged at 14,000 rpm for 10 min. Two hundred μL of samples were placed into a microplate, and the absorbance was measured at 630 nm in a microplate spectrophotometer (Multiskan SkyHigh. Thermo Scientific, Waltham, MA, USA). O_2_^•−^ was indirectly quantified by a linear regression of the mean NBT absorbance values in samples compared to an H_2_O_2_ standard curve (0, 3, 6, 9, 12, 18, 24, and 30 μM).

### 4.4. H_2_O_2_ Quantification 

We used the Peroxidase Assay Kit (Sigma Aldrich, MAK311). This assay utilizes the chromogenic Fe^3+^−xylenol orange reaction in the mixture. A purple complex is formed when Fe^2+^ is oxidized to Fe^3+^ by peroxides present in the sample that can be measured at 585 nm, and the absorbance is proportional to the peroxide level in the sample. The optimized formulation reduces interference by substances in the raw samples. A concentration of 1 μM H_2_O_2_ equals 34 ng/mL, with a H_2_O_2_ range detection between 0.2–30 μM (7–1020 ng/mL). Samples (50 mg) were ground with liquid nitrogen, and 0.5 mL of ultrapure water were added. Samples were mixed for 1 min and centrifuged at 14,000 rpm for 15 min at 4 °C. The supernatant was recovered for the assay, and a 40 µL sample was added to 200 µL of detection reagent in a 96-well plate. After 30 min of incubation, the absorbance was measured as before. The H_2_O_2_ contents in the samples were calculated from a standard curve (0, 3, 6, 9, 12, 18, 24, and 30 μM).

### 4.5. Jasmonic Acid Quantification

Ground tissues (250 mg) were mixed with 1 mL of ethyl acetate and 10 µL of 0.1 mg/mL (±)-Dihydrojasmonic acid (internal standard; Cat: CDS022683, Sigma-Aldrich) and incubated at 4 °C in darkness overnight. The mixtures were centrifuged at 14,000 rpm for 15 min at 4 °C. Supernatants were collected and placed in a 1.5 mL tube. Additionally, plant residues were re-extracted with 1.0 mL of ethyl acetate, centrifugated as before, and the new supernatants were combined, evaporated, and concentrated entirely on a vacuum concentrator. Samples were derivatized with 100 µL of N′N′-diisopropylethylamine, 100 µL of chloroform, and 10 µL of pentafluorobenzyl at 65 °C in an extraction hood. The mixtures were cooled down on ice and were concentrated with gaseous nitrogen again. Finally, the residues were resuspended in methanol (70%) and centrifuged at 14,000 rpm for 5 min at 4 °C. Subsequently, the supernatants were used to analyze JA in a GC-MS (Agilent Technologies, Santa Clara,, CA, USA) equipped with a DB-1MS column (approx. 60 m length, 0.25 mm diameter, and 0.25 μm film) (Agilent Technologies, USA) with the following program 150 °C for 3 min and ramped up at 4 °C/min to 260 °C, with a final temperature after 25 min. The quantification of the JA was performed using the internal standard of dihydrojasmonic added to each sample, as indicated above. An average of the areas of the internal standard was obtained for all the samples, obtaining a correction factor by dividing the area of the internal standard by the area of each sample and multiplying it by the correction factor obtained for each sample. The average area of the internal standard corresponds to a concentration of 0.005 μg/μL [122].

### 4.6. RNA Extraction from Mesquite Leaves Treated with exDNA

Samples treated with exDNA were homogenized in liquid nitrogen with a mortar and pestle to obtain a fine powder. Next, 1 mL of Trizol^®^ reagent (Invitrogen, Waltham, MA, USA) was added before the samples thawed. Samples were mixed by immersion and incubated for 5 min at room temperature. After this time, 200 μL of chloroform were added and vortexed vigorously for 15 s, incubated for 3 min, and centrifuged at 10,000 rpm for 5 min. The aqueous phases were transferred to new 1.5 mL tubes, and 0.5 mL of isopropanol were added. The mixtures were incubated for 10 min at room temperature and centrifuged at 10,000× *g* for 10 min. The aqueous phase was removed from the samples, and the pellets were washed with 1 mL of 70% ethanol. Mixtures were then centrifuged at 10,000 rpm for 3 min, the supernatant was discarded, and pellets were air-dried for 5 min. RNA pellets were then dissolved in 50 μL DEPC–water. The integrity of the RNAs was verified at 230, 260, and 280 nm (the A260–A230 and A260–A280 ratios were calculated), as well as by electrophoresis on agarose gel (1.5%) [123].

### 4.7. cDNA Synthesis and Analysis of MAPKs Gene Expression 

cDNA synthesis was performed using the oligo primer (dT)20 and the SuperScript III Reverse Transcriptase cDNA synthesis kit (Invitrogen^®^). Briefly, an aliquot of isolated RNA was mixed with 1 μL of Oligo dT20 + 1 μL of 10 mM dNTPs mix in a final volume of 10 μL with DEPC water. The mixtures were incubated at 65 °C for 5 min, and subsequently placed on ice for 1 min. The mixture for cDNA synthesis was prepared as follows: 2 μL of 10× reverse transcription buffer, 4 μL of 25 mM MgCl_2_, 2 μL of 0.1 M DTT, 1 μL of RNAase OUT, and 1 μL SuperScript III RT^®^. The mixtures were incubated for 50 min at 50 °C, and the reaction was finished at 85 °C for 5 min and placed on ice. One μL of RNase H was added to the mixture and incubated for 30 min at 37 °C, followed by 1 μL of RNAase A for 30 min. The cDNA obtained was subjected to RT-PCR.

Relative expression levels of *MAPK2*, *MAPK4,* and *MAPK10* genes were quantified by semiquantitative RT-PCR using specific primers (Appendix A). Those primers were previously derived from the oligonucleotide sequences for the *MAPKs* genes that were obtained as a BLAST resulted in sequence comparison from *Prosopis cineraria (*www.ncbi.nlm.nih.gov/datasets/genome/?taxon=364024; accessed on 8 may 2023) and *Prosopis alba* (www.ncbi.nlm.nih.gov/datasets/genome/GCF_004799145.1/; accessed on 9 may 2023) genomes that are annotated in the NCBI databases (www.ncbi.nlm.nih.gov/; accessed 9 May 2023). We used a PCR volume that contained 2.5 μL 10× DreamTaq ^®^ buffer (Thermo Scientific), 1 μL (1 U) DreamTaq ^®^ enzyme (Thermo Scientific), 1 μL cDNA, 0.5 μL (2.5 mM) dNTPs, 1 μL (5 mM) of each primer, and DNase. RNase-free distilled water was used to obtain a final volume of 25 μL. The amplification parameters consisted of 1 cycle at 94 °C for 3 min, followed by 30 cycles at 94 °C for 30 s, 60 °C for 30 s, 72 °C for 30 s, and 1 cycle at 72 °C for 7 min. Mixtures were placed on a 96-well PCR T100 thermal cycler (Bio-Rad, Hercules, CA, USA). The transcription level was determined using *ACTIN1* gene as endogenous control and normalized with the respective control of each treatment. A densitometric analysis was carried out using ImageJ software (accessed on September 2023), (Rasband, W.S., ImageJ with 64-bit Java 8, U. S. National Institutes of Health, Bethesda, Maryland, USA, https://imagej.net/ij/, 1997–2018).

### 4.8. Quantification of Polyphenols and Flavonoids

Lyophilized samples (50 mg) were placed in a 2 mL microtube, mixed with 500 µL of 70% methanol, and homogenized at 1400 rpm in darkness at room temperature overnight. Samples were centrifuged at 14,000 rpm for 15 min to obtain the methanolic extract. Three µL of the sample extract were diluted with 237 µL of deionized water, 15 µL of 10% Folin–Ciocalteu, and 45 µL of 15% Na_2_CO_3_ on a 96-well plate. The mixtures were incubated in darkness at room temperature for 30 min to read the absorbance at 760 nm (Multiskan SkyHigh, Thermo Scientific). Polyphenol concentrations were indirectly determined by linear regression of the mean absorbance values in samples compared to a standard curve. The standard curve was prepared with known concentrations of gallic acid dissolved in deionized water. Flavonoids were also determined according to [124] with some modifications; 12.5 µL of the methanolic extract were mixed with 7.5 µL of 5% NaNO_2_ and incubated at 4 °C for 5 min. Fifteen µL of 10% AlCl_3_.6H_2_O were added, mixed, and incubated for 5 min. Finally, 50 µL of 1M NaOH and 165 µL of deionized water were added to read the absorbance at 510 nm (Multiskan SkyHigh Thermo Scientific). The flavonoid concentration was indirectly determined by linear regression of the mean absorbance values in the samples compared to a standard curve. The standard curve was prepared with known concentrations of quercetin dissolved in deionized water.

### 4.9. Antioxidant Enzyme Activities

Leaves from each treatment were ground with liquid nitrogen for protein extraction; ~100 mg of ground tissue were homogenized with 500 µL of 50 mM Na_2_PO_4_ pH = 7, 1 mM EDTA, and 1% PVP. Next, the homogenates were centrifuged at 4 °C and 14,000 rpm for 15 min. The protein concentration extracted was estimated by Bradford prior to the enzymatic assays. SOD activity was determined according to Méndez-Gómez et al. [125] with some modifications. Briefly, in 1.5 mL microtubes, 10 µL of extract were mixed with 40 µL of 0.1 M EDTA, 20 µL 1.5 mM NBT, 600 µL of 0.0067 M KH_2_PO_4_ pH = 7, and 50 µL of 12 mM of riboflavin. Reaction control is the same except for samples. The microtubes were illuminated with a 40 W lamp for 15 min. Posteriorly, absorbance at 560 nm was measured every 1 min for 10 min (Multiskan SkyHigh Thermo Scientific). For calculations, the blank absorbance was subtracted from each reading made at 560 nm on the samples that were tested. Activity was defined as the amount of enzyme that inhibits a change of 0.001 in absorbance caused by the photochemical reduction of NBT. For PEX, 140 µL of 100 mM Na_2_PO_4_ pH = 6, 50 µL of 5 mM guaiacol, 10 µL of extracted, and 10 µL of 5 mM H_2_O_2_ were mixed in a 96-well plate to read the absorbance at 480 nm every min for 30 min (Multiskan SkyHigh, Thermo scientific, Waltham, MA, USA). The activity was defined as the amount of enzyme that increases by 0.001 in absorbance by guaiacol oxidation. CAT activity was determined by a decrease in H_2_O_2_ concentration [126]; 2.5 µL of extract were mixed with 150 µL of 50 mM Na_2_PO_4_ pH = 7 and 20 µL of 50 mM H_2_O_2_ in a 96-well plate to read the absorbance at 240 nm each min for 10 min (Multiskan SkyHigh, Thermo scientific, Waltham, MA, USA). The activity was defined as the amount of enzyme that disproportionates 1 μmol H_2_O_2_ for 1 min. The H_2_O_2_ extinction coefficient used for the calculation was ε = 45.2 /mM mL.

### 4.10. Phytohormone Quantification

The phytohormones were extracted and purified in accordance with previous reports by Zárate-López et al. [54]. Samples were suspended in 250 µL of water of HPLC grade that were acidified at 0.1% with acetic acid (J. T. Baker, Phillipsburg, NJ, USA). Chromatographic analysis was performed on a UHPLC Ultimate 3000 (Thermo Scientific, San Jose, CA, USA) equipped with a quaternary pump, diode array detector (DAD), and a Zorbax RRHD C18 column (150 mm × 2.1 mm id, and 1.9 µm of particle size; Agilent Technologies, San Jose, CA, USA). The chromatography elution was done with ternary gradient elution combining methanol (B), acetonitrile (C), and water supplemented with 0.1% acetic acid (A) at a flow rate of 0.35 mL/min. The chromatography elution was done with ternary gradient elution combining methanol, acetonitrile, and water supplemented with 0.1% acetic acid at a flow rate of 0.35 mL/min. The following ternary gradient was used: 100% A (zero to one min), 60% A + 35% B + 5% C (one to 16 min), and 80% B + 20% C (16 to 21 min). A UV-DAD detector was set to record spectra between 210 nm and 270 nm. The quality control of the sample developed in a linear range between 0.98 to 35.91 uM of phytohormones. Calibration graphs were generated by plotting the peak areas against the corresponding concentrations injected into the HPLC column. Least-squares linear regression analysis was used to generate a linear calibration curve. The limit of detection (LOD) and limit of quantification (LOQ) were established as 3 times and 10 times, respectively, the signal–noise ratio. The HPLC chromatograms were analyzed using Chromeleon 7.0 software (Thermo Scientific, San Jose, CA, USA).

### 4.11. Statistics Analysis

For all experiments, the data were statistically analyzed using STATISTICA 8.0 Software (Dell StatSoft, Austin, TX, USA). Multivariate analyses with Tukey’s post hoc test were used for testing differences in the analyzed parameters. In the graphs, the different letters indicate significant differences in comparison to the control (*p* ≤ 0.005).

## 5. Conclusions

In this study, we analyzed the stress responses to the exogenous application of self exDNA and nonself exDNA from the mistletoe *P. calyculatus* as a source of DAMPs in mesquite trees (*P. laevigata*). The study showed that the tree exhibits comparable reactions to both self and nonself exDNA applications, and both types of exDNA sources induced differential species-specific responses, such as ROS generation, elevated levels of defense metabolites like polyphenols and flavonoids, ROS-stress enzyme activities that reduce the oxidative burst, and the spatial synthesis of phytohormones. Furthermore, we observed differential expression in genes encoding MAPKs mainly when self exDNA was applied. Like other organisms, mesquite trees respond with self exDNA-inducing responses, such as oxidative burst and MAPKs activation. However, we believed that, for their ecological relation, mesquite trees respond to mistletoe exDNA with weaker responses compared to their self exDNA. 

## Figures and Tables

**Figure 1 ijms-25-00457-f001:**
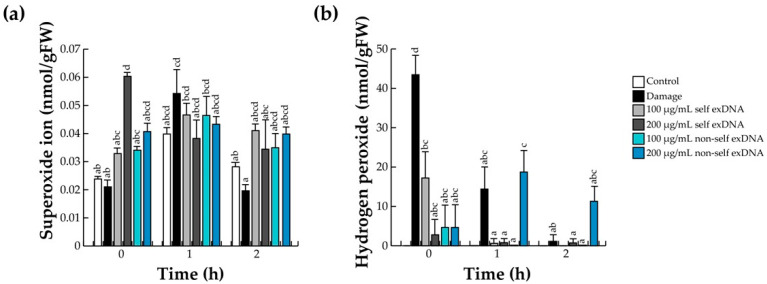
Quantitative determination of ROS in mesquite leaves treated with exDNA sources. Concentrations of O_2_^•−^ (**a**) and H_2_O_2_ (**b**). Mesquite leaves were wounded (damage) or sprayed with water for control, or self-exDNA (derived from mesquite leaves), and non-self exDNA (derived from *P. calyculatus* mistletoe leaves), and then collected at 0, 1, and 2 h. Bars represent means ± SE (*n* = 5; nmol per g fresh weight), and the different letters indicate a significant difference at *p* ≤ 0.05.

**Figure 2 ijms-25-00457-f002:**
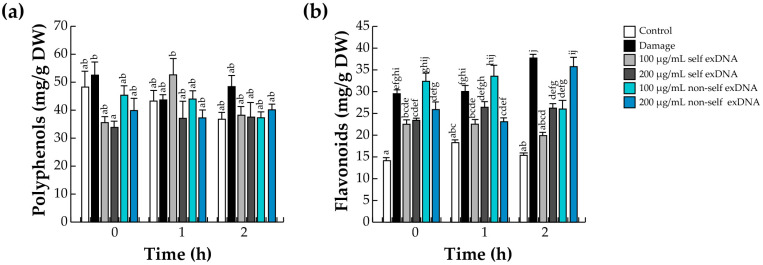
Polyphenol and flavonoid quantification in mesquite leaves treated with exDNA sources. Polyphenol (**a**), and flavonoid (**b**) levels. Mesquite leaves were wounded (damage) or sprayed with water for control, or self-exDNA (derived from mesquite leaves), and non-self xDNA (derived from *P. calyculatus* mistletoe leaves), and then collected at 0, 1, and 2 h. Bars represent means ± SE (*n* = 5; milligrams per g of dry weight: DW), and the different letters indicate a significant difference at *p* ≤ 0.05.

**Figure 3 ijms-25-00457-f003:**
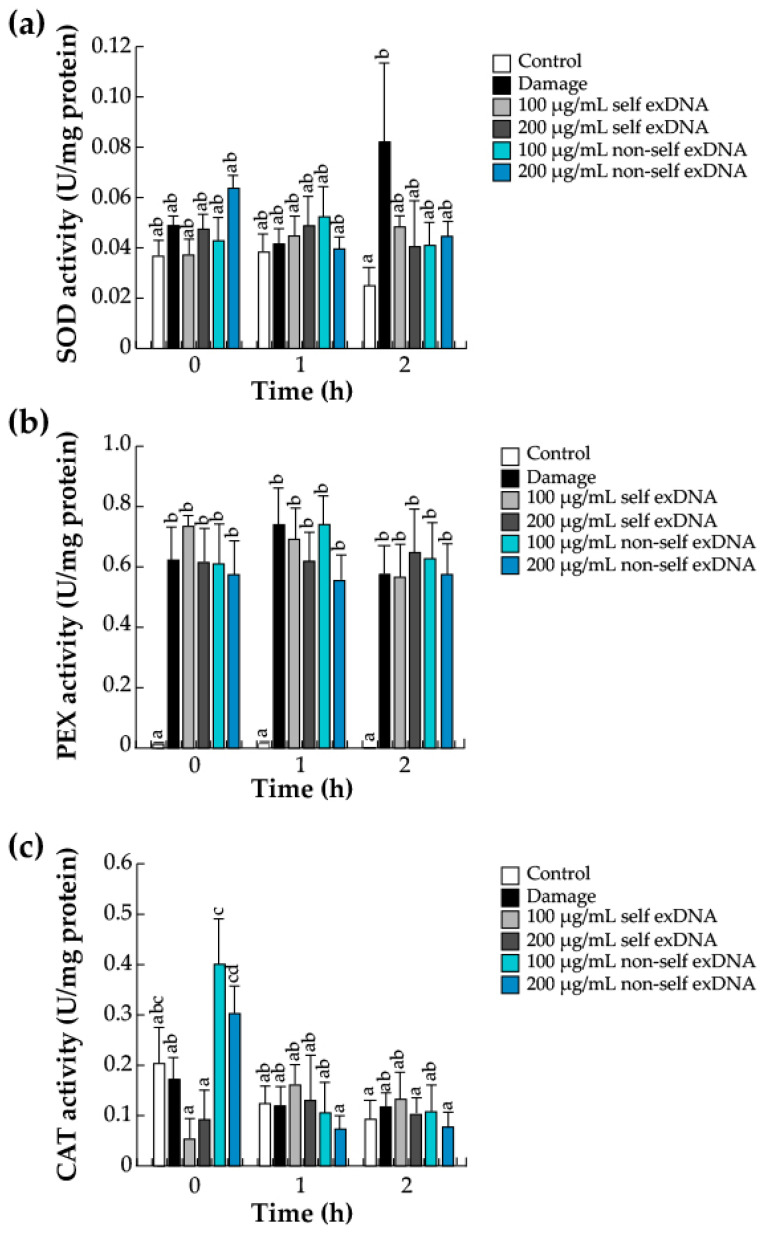
Antioxidant enzyme activities in mesquite leaves treated with exDNA sources. Superoxide dismutase (SOD) (**a**), peroxidase (PEX) (**b**), catalase (CAT) (**c**) activities. Mesquite leaves were wounded (damage) or sprayed with water for control, self-exDNA (derived from mesquite leaves) and non-self exDNA (derived from *P. calyculatus* mistletoe leaves), and then collected at 0, 1, and 2 h. Bars represent means ± SE (*n* = 5; units of activity per mg of protein), and the different letters indicate a significant difference at *p* ≤ 0.05.

**Figure 4 ijms-25-00457-f004:**
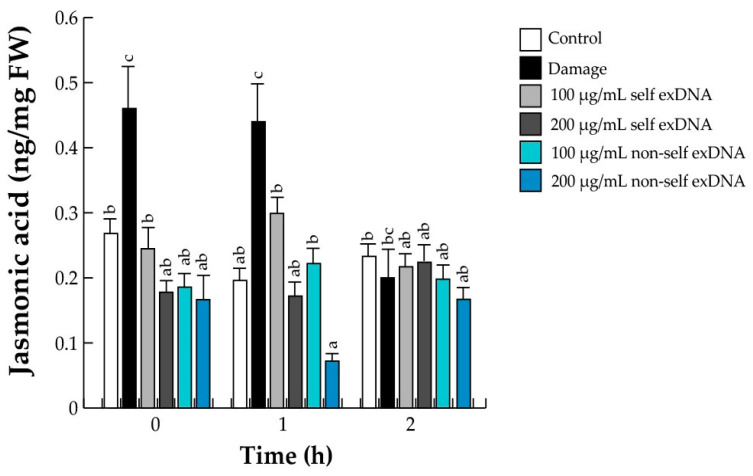
Quantitative JA determination in mesquite leaves treated with exDNA sources. Mesquite leaves were wounded (damage) or sprayed with water for control, self-exDNA (derived from mesquite leaves), and non-self exDNA (derived from *P. calyculatus* mistletoe leaves), and then collected at 0, 1, and 2 h. Bars represent means ± SE (*n* = 5; nanograms per mg of fresh weight), and the different letters indicate a significant difference at *p* ≤ 0.05.

**Figure 5 ijms-25-00457-f005:**
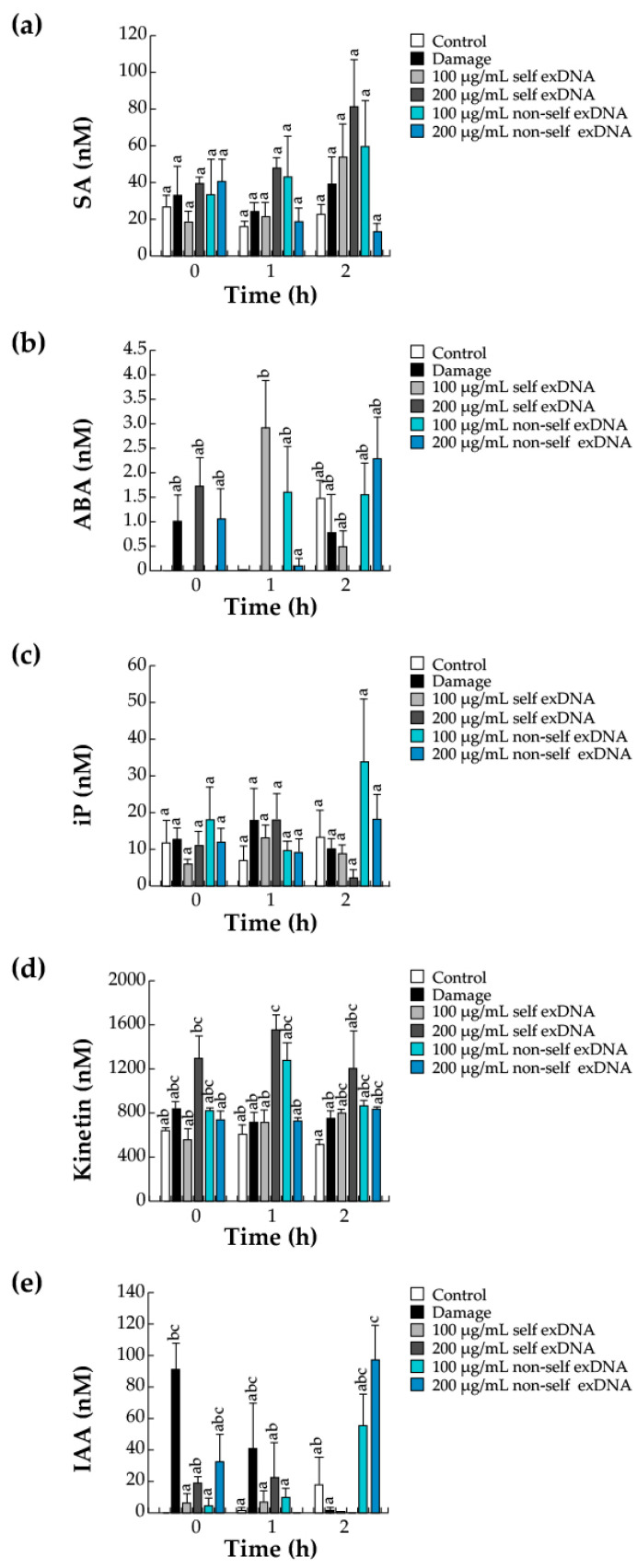
Quantitative phytohormone determinations in mesquite leaves treated with exDNA sources. Salicylic acid (SA) (**a**), abscisic acid (ABA) (**b**), N6-(2-isopentenyl)adenine (iP) (**c**), 6-Furfurylaminopurine (Kinetin) (**d**), and 3-indoleacetic acid (IAA) (**e**) contents. Mesquite leaves were wounded (damage) or sprayed with water for control), or self-exDNA (derived from mesquite leaves) and non-self exDNA (derived from *P. calyculatus* mistletoe leaves), and then collected at 0, 1, and 2 h. Bars represent means ± SE (*n* = 5), and the different letters indicate a significant difference at *p* ≤ 0.05.

**Figure 6 ijms-25-00457-f006:**
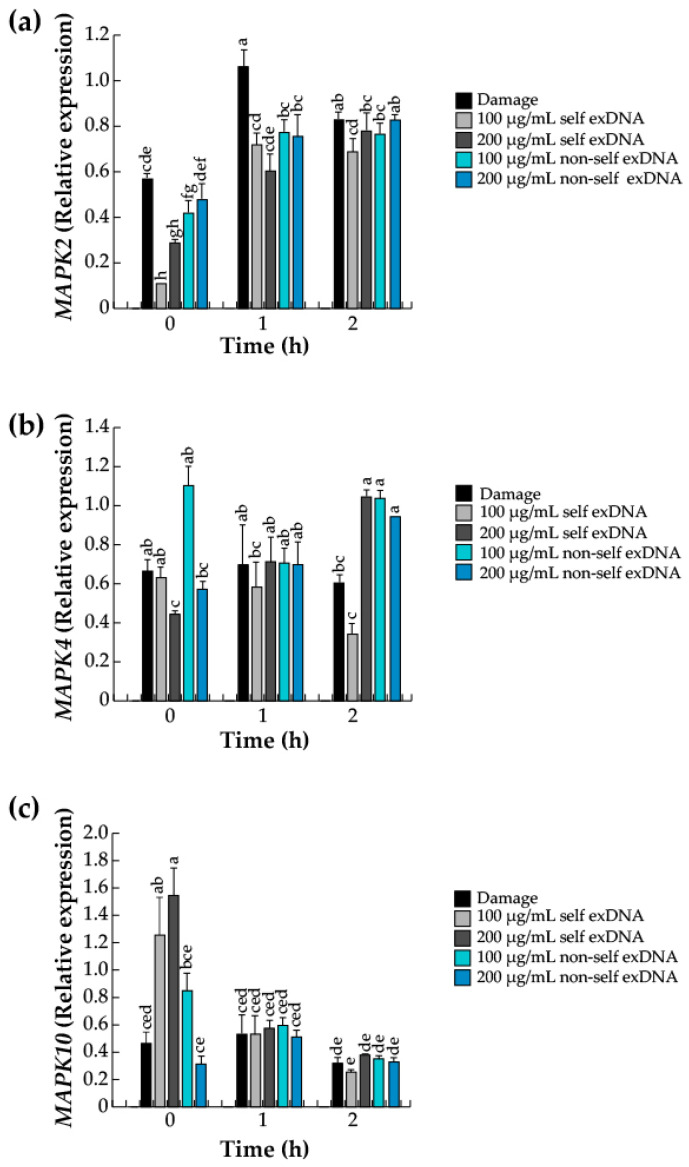
exDNA sources induce differentially relative gene expression of *MAPKs* in mesquite leaves. *MAPK2* (**a**), *MAPK4* (**b**), and *MAPK10* (**c**) gene expressions. Mesquite leaves were wounded (damage) or sprayed with water for control, self-exDNA (derived from mesquite leaves) and non-selfexDNA (derived from *P. calyculatus* mistletoe leaves), and then collected at 0, 1, and 2 h. Bars represent means ± SE (*n* = 5), and the different letters indicate a significant difference at *p* ≤ 0.05.

**Figure 7 ijms-25-00457-f007:**
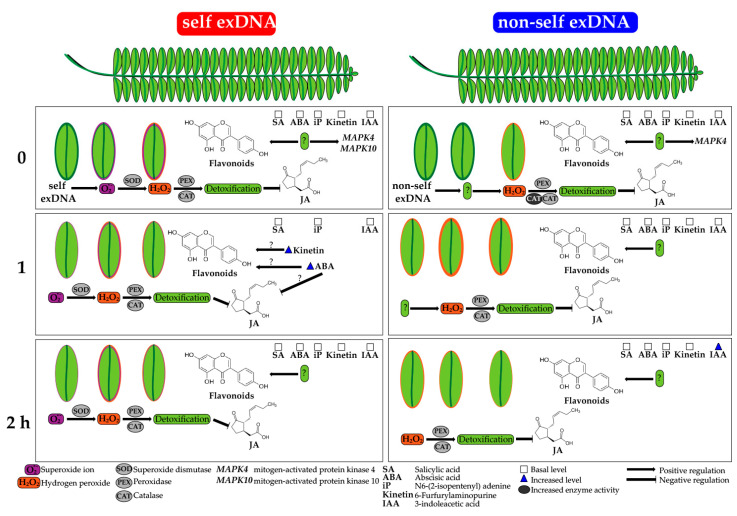
Responses of *P. laevigata* to the exogenous exDNA applications. O^2•−^ accumulation in the leaf represented by violet coloration, H_2_O_2_ accumulation in the leaf represented by red coloration, arrows = induction, perpendicular lines = inhibition, white squares = basal level of phytohormones and blue triangles = increase in phytohormone content (See details in text).

## Data Availability

Data are contained within the article and Appendix A.

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
