# Peer review of "Extracellular Self- and Non-Self DNA Involved in Damage Recognition in the Mistletoe Parasitism of Mesquite Trees"

_ijms, 2023, doi:10.3390/ijms25010457_

Round 1
Reviewer 1 Report
Comments and Suggestions for Authors
The manuscript sheds light into the response of mesquite to self- and ex-DNA. The study of the responses is complete and covers a wide range of genetic, biochemical, enzimatic and hormonal responses, identifying patterns and differences between treatments. I think that the results from the authors are presented in a clear and organized way and the conclusions of the study are supported by the results.
Line 28: is it a Mexican endemic species? There is evidence that this species occurs in other countries in South America.
Results 2.1: Check species names in italics
Discussion: Check species names in italics
Line 280: “S. lycopersicum” is misspelled.
Line 305: “S. lycopersicum” is misspelled.
Line 395: the host is referred to twice in the same sentence as host and as P. laevigata.
Discussion: Perhaps the discussion is missing a brief part where ideas of how to expand the knowledge in the area should advance in the future. Future studies using susceptible mesquite trees and tolerant ones (e.g. trees with no signs of parasitism in a heavily parasitized population, what other genes to test, insights into MAPK specificity, etc.).
General comment: I would like to ask the authors their opinion as to what if the ex-DNA in the study would have been from a tomato, for example. Would the results have been the same/equivalent? In this sense, it would need to be stressed the difference of using ex-DNA from a parasitic species and a non-parasitic species.
Author Response
The manuscript sheds light into the response of mesquite to self- and ex-DNA. The study of the responses is complete and covers a wide range of genetic, biochemical, enzymatic and hormonal responses, identifying patterns and differences between treatments. I think that the results from the authors are presented in a clear and organized way and the conclusions of the study are supported by the results.
Response: Thank you for your mention to our work.
Line 28: is it a Mexican endemic species? There is evidence that this species occurs in other countries in South America.
Response: Mesquite (Prosopis laevigata) is an important tree and northern Mexico. This tree is native to Mexico, however, the species is also found in in the arid southwestern United States, and Peru. In addition, Prosopis genus is widely distributed in the dry regions of America such as Belice, Argentina, and certain species can be confused. According to the authors (check the citation here), they mention that this particular species can be found in high concentration in the Guanajuato Bajio area in center of Mexico.
Villalón-Mendoza, H., Hernández-Hernández, E. E., & Manzanares-Miranda, N. (2023). Presence and Importance of Mesquite Prosopis laevigata (Humb. & Bonpl. ex Willd.) MC Johnst in Northeastern Mexico. In Sustainable Management of Natural Resources: Diversity, Ecology, Taxonomy and Sociology (pp. 115-129). Cham: Springer International Publishing.
https://www.fao.org/3/q2580e/q2580e07.htm
We added a sentence in Page (P): 1, Line (L):31
Results 2.1: Check species names in italics
Response: The suggestion was corrected
Discussion: Check species names in italics
Response: The suggestion was corrected
Line 280: “S. lycopersicum” is misspelled.
Response: The suggestion was corrected
Line 305: “S. lycopersicum” is misspelled.
Response: The suggestion was corrected
Line 395: the host is referred to twice in the same sentence as host and as P. laevigata.
Response: The sentence was corrected. Check Page: 18, Lines: 517-519.
Discussion: Perhaps the discussion is missing a brief part where ideas of how to expand the knowledge in the area should advance in the future. Future studies using susceptible mesquite trees and tolerant ones (e.g. trees with no signs of parasitism in a heavily parasitized population, what other genes to test, insights into MAPK specificity, etc.).
Response: Thank you for analyzing our research and noting this critical fact. In the discussion section, we have enriched the discussion section. You can see it: Page (P): 15; Lines (L): 370-383; P:15, L:385-391; P:16, L:406-412; P:16, L:426-432; P:17, L:468-478; P:17, L:493-502; P:18, L:508-517; P:18, L:526-530.
General comment: I would like to ask the authors their opinion as to what if the ex-DNA in the study would have been from a tomato, for example. Would the results have been the same/equivalent? In this sense, it would need to be stressed the difference of using ex-DNA from a parasitic species and a non-parasitic species.
Response: Based on reports, we expect that the responses of mesquite using exDNA from a non-parasitic and taxonomically distant plant will be lower compared to DNA mistletoe responses because the response to extracellular DNA is species-specific (non-self-DNA from plant species that does not cause disease or parasitism), Furthermore, under natural conditions the tomato will never be in contact with the mesquite. However, the mesquite is in constant interaction with the mistletoe, so we hypothesize that the mesquite has adapted defense responses against the DNA, mRNA, effectors, peptides, and fragments of the cell wall of a species that causes direct damage to it. It is known that responses to extracellular DNA are species-specific, that is, the response to DNA decreases as the source DNA becomes taxonomically distant from the DNA of the recipient species, but these responses have only been tested with non-species extracellular DNA, that do not cause diseases in the recipient species, for example: applying salmon DNA on Arabidopsis, or Arabidopsis on any plant species. There are few articles where non-pathogen exDNA is used, and the observation is that the recipient species responds more to pathogen exDNA than to the control with distilled water, which is completely different from what was observed with non-pathogen exDNA where the response to this type of extracellular DNA is almost null, with a response almost identical to the controls with distilled water.

Reviewer 2 Report
Comments and Suggestions for Authors
The paper delves into a novel and intriguing area of research by investigating the role of extracellular self- and non-self-DNA in mistletoe parasitism on mesquite trees. This approach contributes to the understanding of plant interactions. The data presentation is well-organized, and the findings are supported by robust data. However, several issues should be addressed and hence the paper may be a
1- Several sections of the m/s are convoluted, making it challenging to follow the arguments. A revision for clarity and coherence would enhance the overall readability of the paper.
2- The paper lacks a comprehensive discussion of potential limitations in data collection or analysis. Identifying and addressing any biases or confounding factors would strengthen the validity of the results.
3- The discussion and conclusion sections could benefit from a more in-depth exploration of the implications of the results. A more robust connection between the findings and their broader implications within the field would enhance the overall impact of the paper. Also, there are instances where the connection between the current study and prior research could be strengthened.
4- Reasons why researchers choose to investigate MAPKs in the context of mistletoe parasitism on mesquite trees?.
5- A (metabolic) connection among all the measured parameters is needed in the text, instead of keeping them isolated.
6- Other drawback is The timing of sample collection (0, 1, and 2 hours) might be relatively short for certain aspects, especially considering the potential time-dependent nature of plant responses. The rationale for the chosen time points could be explained more thoroughly.
7- Although it's mentioned that genomic DNA and exDNA generation were verified on agarose gel, the gel results are not clear. Authors should strengthen the validation process.
8- M&M section: The quantification methods for O2•−, H2O2, and jasmonic acid are briefly described. Providing more details on the calibration curves, sensitivity, and specificity of these assays would enhance the robustness of the reported results.
Comments on the Quality of English Language
Extensive editing of English language required
Author Response
The paper delves into a novel and intriguing area of research by investigating the role of extracellular self- and non-self-DNA in mistletoe parasitism on mesquite trees. This approach contributes to the understanding of plant interactions. The data presentation is well-organized, and the findings are supported by robust data. However, several issues should be addressed and hence the paper may be a
Response: Thank you for your mention to our work.
1- Several sections of the m/s are convoluted, making it challenging to follow the arguments. A revision for clarity and coherence would enhance the overall readability of the paper.
Response: Thank you for analyzing our research and noting this critical fact. We have reanalyzed all manuscript sections to improve understanding and added contexts in each part (Introduction, results, and discussion). You can check different yellow flags that we added in the text. Additionally, we removed Supplementary Figures S3 and S4 for better visibility. It involves the application by spraying the exDNA to the branches (S3) and the geolocation points of the trees treated with the exDNA (S4), described in the materials and methods.
2- The paper lacks a comprehensive discussion of potential limitations in data collection or analysis. Identifying and addressing any biases or confounding factors would strengthen the validity of the results.
Response: Thank you for analyzing our research and noting this critical fact. In the discussion section, we have enriched the discussion section. You can see it: Page (P): 15; Lines (L): 370-383; P:15, L:385-391; P:16, L:406-412; P:16, L:426-432; P:17, L:468-478; P:17, L:493-502; P:18, L:508-517; P:18, L:526-530.
3- The discussion and conclusion sections could benefit from a more in-depth exploration of the implications of the results. A more robust connection between the findings and their broader implications within the field would enhance the overall impact of the paper. Also, there are instances where the connection between the current study and prior research could be strengthened.
Response: Thank you for analyzing this part and noting this critical fact. In the discussion section, we have added connections between sections. You can see it: Page (P): 15; Lines (L): 370-383; P:15, L:385-391; P:16, L:406-412; P:16, L:426-432; P:17, L:468-478; P:17, L:493-502; P:18, L:508-517; P:18, L:526-530.
4- Reasons why researchers choose to investigate MAPKs in the context of mistletoe parasitism on mesquite trees?
Response: Thank you for analyzing this part and noting this critical fact. We have added part of this text in P:17, L:493-502
Based on the reports, in the infective process, organisms require an endogenous signaling pathway that enables them to perceive injury through DAMPs, to mount adequate local and systemic responses to activate the immune system and achieve resistance. DAMPS that trigger an immune response are ATP, extracellular matrix fragments, and extracellular DNA (exDNA), which activate signaling minutes after the perception of damage with events such as calcium fluxes in the cytosol, ROS, and phosphorylation of mitogen-activated protein kinases (MAPKs) which are considered conserved early responses. Related to the mistletoe and host tree, an intrusive process (mediated by the haustoria penetration) produces physical damage, and the host can perceive this damage by releasing different DAMPs and expressing general defense responses or activating specific innate immune response cascades to fend off parasitic ingress. In that aspect, the level of expression of genes that coding for MAPKs in the infective process among parasitic plants, since host plants can respond to the infective process and perceive the damage through the regulatory process through MAPKs and the hosts can be able to respond, through oxidative stress. We are conducting pilot tests among young mesquite trees to analyze the perception and defensive responses against the mistletoe.
5- A (metabolic) connection among all the measured parameters is needed in the text, instead of keeping them isolated.
Response:Thank you for analyzing this part and noting this critical fact. We have added that important part in: P:16, L:426-431, and P:17, L:468-478; P:18, L:508-517.
6- Other drawback is The timing of sample collection (0, 1, and 2 hours) might be relatively short for certain aspects, especially considering the potential time-dependent nature of plant responses. The rationale for the chosen time points could be explained more thoroughly.
Response: Thank you for analyzing this part and noting this critical fact.
The times are based on the different reports that we have analyzed (Table 1). However, we know the various biomolecules analyzed require more hours or even days to respond to physiological stress (metabolites such as polyphenols and flavonoids or phytohormones). However, according to our results, we obtained stress responses by applying exDNA. Under criticism of starting new research and more in a plant organism such as a tree. We decided to start with these times (0, 1, and 2 h) because these times are consensus and similar in different plants such as beans, Arabidopsis, Rice, grapes, and microalgae, to mention some examples (See Table 1). Our future idea is to address longer analysis times (4, 6, 8 h), and analyze different biomarkers.
We added some details in: P.7, L:134-135
7- Although it's mentioned that genomic DNA and exDNA generation were verified on agarose gel, the gel results are not clear. Authors should strengthen the validation process.
Response: Thank you for analyzing this part and noting this critical fact. We have added more description in the agarose gel figure S2.
8- M&M section: The quantification methods for O2•−, H2O2, and jasmonic acid are briefly described. Providing more details on the calibration curves, sensitivity, and specificity of these assays would enhance the robustness of the reported results.
Response: Thank you for analyzing this part and noting this critical fact. We have added more specifically details to each of the methodologies.
You can check details for O2•− in P: 20; L:599-610. H2O2 in P: 20; L:614-626. and JA in P: 20-21; L:642-645.

Round 2
Reviewer 2 Report
Comments and Suggestions for Authors
I have carefully reviewed the revised m/s, and I am pleased to acknowledge the substantial improvements made in response to the initial feedback. The authors have diligently addressed all the concerns raised during the first round of review.
Based on the thorough revisions and improvements made by the authors, I am pleased to recommend the acceptance of the m/s for publication.
I look forward to seeing the m/s in print.
Congratulations !!
Minor editing of English language required